# Interface Design for Products for Users with Advanced Age and Cognitive Impairment

**DOI:** 10.3390/ijerph19042466

**Published:** 2022-02-21

**Authors:** Li-Hao Chen, Yi-Chien Liu

**Affiliations:** 1Department of Applied Arts, Fu Jen Catholic University, New Taipei City 24205, Taiwan; 2Neurology Department, Cardinal Tien Hospital, New Taipei City 23148, Taiwan; milkgen@gmail.com

**Keywords:** advanced age, dementia, focus group, intuitive interaction, mild cognitive impairment (MCI)

## Abstract

The aim of this study is to investigate the problems and needs of users with advanced age and cognitive impairment regarding the design and operation of daily living products. Television remote controls and an electric rice cooker were applied as the research tools, and focus group interviews with control older adults and interviews with individuals with MCI or mild dementia were conducted regarding the operation of the products. The control participants stressed that the operating procedures should not be excessively complex, the number of functions and buttons should not be overly high, and buttons and text should be enlarged. For those with MCI or mild dementia, in addition to the size and number of buttons, text size, and functions, their operation of product interfaces was affected by the complexity of the operating procedures. The solutions recommended by the participants included interface design involving direct operation and voice control.

## 1. Introduction

Older adults now attract the attention of designers. Factors affecting the daily living of older adults, in addition to the decline in personal cognition and motor functions with age, include dementia. Dementia affects the cognitive abilities and short-term memory of many older adults [1,2], along with daily living activities such as social and physical activities [2]. Notably, mild cognitive impairment (MCI) is considered an early symptom of dementia. MCI is a transitional stage from general aging to dementia. With aging populations, the prevalence of MCI has increased substantially [3]. MCI exacerbates into dementia in 10–15% of individuals, which constitute 1–2% of the population of older adults [4]. Subjective cognitive decline is a prevalent symptom among individuals with MCI; they perceive their memory as having declined even though examinations indicate that their memory is normal. Specifically, they exhibit signs of memory decline but not dementia, and their daily living behaviors and capabilities have not been severely affected; only abilities required for complex daily living activities have been impeded [5], such as using technological products, cooking, and taking medication. Daily living technologies have become critical for self-care and daily living activities [6], and MCI may hamper their use [7], negatively affecting independent living at home and increasing the burden on nursing homes [6]. Individuals with MCI commonly overlook the importance of treatment because of their lack of insight, resulting in mild dementia. Mild dementia is regarded as the decline in memory and attention and the inability to remember recent or familiar things and occurrences due to age. Individuals with mild dementia exhibit normal mobility and are able to manage most of their daily living activities. Individuals with MCI or mild dementia require attention from the designers of products for older adults because most are still capable of self-care or still live with family. Understanding the needs of individuals with MCI or mild dementia using daily living products is mandatory for grasping the characteristics and requirements of interface designs for them.

Understanding user needs is critical to contemporary product design. In addition to appearance, designers must pay attention to the interactivity and usability of products. According to Courage and Baxter [8], products should adapt to users rather than vice versa. Intuitive interactions have garnered widespread attention in product designs in recent years. Studies have indicated that users’ experience and knowledge play vital roles in intuitive interactions with products. Decline in short-term memory and cognitive abilities caused by dementia renders intuitive interactions particularly challenging for users. Nevertheless, design can improve the quality of life of individuals with mild dementia [9]. Specifically, new product designs can enhance the quality of life of these individuals and their cohabitants [10]. Studies on human–machine interaction have considered designing for users with dementia [11]. A product design that satisfies the needs of users with advanced age and dementia improves their quality of life and dignity and reduces the labor burden in medical and home care.

Cognitive impairment affects the interaction of users with advanced age with product interfaces. It is worth of further investigating whether older adults with cognitive impairment differ from normal older adults in their interaction with product interfaces. Enabling individuals with MCI or mild dementia to experience technology product operation is a feasible means to understanding their design needs. Therefore, the purpose of this study is to explore the problems encountered by older adults with MCI or mild dementia in operating the interfaces of daily living products to clarify the product interface designs that satisfy their needs.

## 2. Literature Review

Designs focusing on users with MCI or dementia have recently attracted attention in the field of design. According to Irizarry et al. [12], collaborating with individuals with dementia in interface design poses a considerable challenge. Nevertheless, recent studies have investigated technological product interface design requirements for users with MCI or dementia from the perspectives of interface operation tasks or processes, appearance, and usage [6,9,13,14]. Schmidt and Wahl [6] explored the performance of 39 control older adults and 41 older adults with MCI in performing specific tasks with three types of daily living products (viz., sphygmomanometers, mobile phones, and e-book readers), revealing that the control older adults exhibited a significantly lower error rate and higher performance than those with MCI. Moreover, longer time was required for the participants with MCI to operate the product interfaces at multiple levels; this significantly affected their interactions with the products.

Hedman et al. [14] conducted a 5-year longitudinal study on the difficulties of 37 users with MCI using daily living products. The results indicated that washing machines, television (TV) remote controls, handheld mixers, and automated teller machines were originally easy for the participants to use but became difficult to use over the course of the study. Compact disc players and digital cameras were already difficult for the participants to use in the first year and became more difficult to operate over the study period. This indicates that cognitive decline or impairment does affect user familiarity with products. To explore the possibility of users with dementia participating in product interface design, Brankaert and Ouden [9] investigated the use of smartphones by 10 patients with dementia (nine were diagnosed as having mild dementia, one as having moderate dementia) and their caregivers. The results revealed that human factors and cognitive problems hampered the patients’ use of the smartphones. Particularly, they experienced difficulty in understanding the functional structure of the main menu, which exhibited excessive features. Moreover, they found the phone interfaces overly complex, even though they had no difficulty reading and understanding the text. In addition, the patients considered the phones excessively large but had difficulty entering text on small screens.

Castilla et al. [15] explored the usability of web page (echoBUTLER digital platform) for users with MCI (see Figure 1). The study invited twenty-eight older adults with MCI as the subjects, and the participants averaged 77.0.8 years (SD = 9.6 years) of age. The results revealed that compared with ordinary users, participants with MCI have a lower awareness of the operation elements of an interface; therefore, primary interactive operation elements should be positioned in the center of the screen. In addition, they tend to press the outer area of a button because they want to know the function of the button they are about the press.

Blackler et al. [16] explored intuitive interaction with various interfaces for people living with dementia. In their study, a usability test of six types of microwave interfaces for adjusting cooking power (see Figure 2) was conducted to examine the relationship between intuitive interactions in product interfaces among older people with dementia. The tests were performed in a hospital clinic. Hospital neurologists provided help to invite the 25 participants who had mild dementia, demonstrated acceptable communicative and cognitive functions, and had experiences in using home appliances. The participants averaged 81.8 years (SD = 7.2 years) of age. The results showed that the indirect user interfaces (i.e., those that require the user to cycle through the settings) required more time to complete. Interface E was more enticing to the participants in their initial encounters with the interface, but the mean time required to complete the tasks using this interface was undesirable. Interface F was the only interface with buttons that were arranged vertically. The mean task completion times attained through this interface were relatively long compared with those achieved through the other interfaces. The study also indicated that user interfaces should be designed in accordance with user prior experiences.

Compared with ordinary older adults, those with MCI differ in terms of their perceptions of the difficulty of using daily technology [17,18]; however, the associated factors (e.g., the complexity of operation tasks and configuration of operation interfaces) have not yet been clarified. Understanding the problems encountered by older adults with MCI and mild dementia in operating the product interfaces can reveal the interface features and design elements that are suitable for them.

## 3. Method

Focus groups and individual interviews were conducted with control older adults, older adults with MCI, and older adults with mild dementia living with family to understand their user needs and the problems they have encountered in using daily living products. Ten control older adults were invited to participate in two focus group interviews with five interviewees. Individual interviews were conducted with eight patients with MCI and seven with mild dementia. After discussion with medical professionals, group interviews were deemed unsuitable for the patients with MCI or mild dementia because emotional volatility may accompany their disease. The following subsection describes the procedure of selecting control participants and those with MCI or mild dementia.

### 3.1. Participants

Control older adults in community centers were invited to participate in the focus group interviews. Participants diagnosed as having MCI or mild dementia were assessed and selected with the assistance of attending physicians.

#### 3.1.1. Selecting Control Participants

Ten older adults that had normal cognitive functions, were experienced in using home appliances and technological products, and exhibited normal verbal communication abilities were selected from two local senior citizen activity centers to participate in focus group interviews. Five older adults were invited from each activity center, with average ages of 81.0 ± 4.7 and 78.0 ± 6.6 years for the groups.

#### 3.1.2. Selecting Participants with MCI

Eight patients with MCI who were aged 77.0 ± 9.2 years, were experienced in using home appliances and technological products, and exhibited normal verbal communication abilities were selected to participate in individual interviews with help from attending physicians. Patients rated 0.5 on the Clinical Dementia Rating Scale were selected as participants with MCI. Through the referral by the physicians, the researchers directly communicated with the patients’ and their families to confirm their willingness to participate in the study.

#### 3.1.3. Selecting Participants with Mild Dementia

Seven patients with mild dementia who were aged 73.1 ± 11.7 years, were experienced in using home appliances and technological products, and exhibited normal verbal communication abilities, were selected to participate in individual interviews with help from attending physicians. Patients rated 1 on the Clinical Dementia Rating Scale were selected as participants with mild dementia. Through the referral by the physicians, the researchers directly communicated with the patients’ and their families to confirm their willingness to participate in the study. Patients with dementia experienced in using home appliances and technological products basic knowledge in product operations; therefore, they encountered few problems in using the products and were able to express opinions regarding these problems.

### 3.2. Focus Group and Individual Interviews

Three types of TV remote controls with different button configurations, sizes, and appearances, and one type of electric rice cooker were employed as the research instruments (Figure 3). The operational tasks involving these two types of test products were set as follows: (1) The rice cookers were used to cook by pressing the “function select” button and then the “cook” button; set an appointed time for cooking by pressing the “function select” button, “appointment time” button, and then “hour/minute setting” button; and set the simmer time by pressing the “function select” button, “minute or cooking time” button, and then the “cook” button. (2) The TV remote controls were used to select channels by pressing the up and down or number buttons and to adjust the volume.

One focus group interview on the products, aimed at older adults, was held at each of the two activity centers in May of 2020. Before the interviews commenced, the principal investigator explained the interview procedure to the participants as well as the functions and usage methods of the rice cooker and remote controls. The participants were asked to perform the aforementioned operations using the two types of products for them to familiarize themselves with and discuss the products. After the tasks were complete, 60-min discussions were held for the participants to express their opinions regarding the use of the products. The discussions were then concluded under the guidance of the principal investigator. Specifically, the rice cooker was discussed first, followed by the remote controls. The tasks and interviews were recorded using audio and video recording equipment for analysis (Figure 4).

The individual interviews with the participants diagnosed as having MCI and mild dementia were held in July and August of 2020. The procedure of the individual interviews was explained to the participants and their families by the principal investigator, along with the functions and usage methods of the rice cooker and remote controls. The participants were then asked to perform the operations. Then, accompanied by family, each participant discussed their use of the products in a 40-min interview. The rice cooker was discussed first, followed by the remote controls. The operations and interviews were recorded using audio and video recording equipment (Figure 5).

### 3.3. Interview Locations and Equipment

The focus group interviews were conducted in social spaces of senior citizen activity centers in New Taipei City and Pingtung City. The individual interviews were held in a meeting room in a hospital in New Taipei City, with the participants selected, assessed, and invited with the help of attending neurologists. To protect data confidentiality, the interview analysis focused on the spoken materials, and the personal information and images of the participants were hidden as much as possible. Digital video cameras, pen voice recorders, and laptop computers were used for recording and analyzing the spoken materials.

## 4. Results

The spoken materials from the interviews were analyzed to generalize the problems regarding the operating procedures, interfaces, functions, and appearances of the test products, which were investigated according to the opinions of the participants regarding the operational tasks, operating interfaces, product functions, and interface appearances, respectively.

### 4.1. Focus Group Interviews

#### 4.1.1. Electric Rice Cooker

The focus group interviews are presented as Sites A and B as shown in Table 1, which lists the focus group interview results regarding the rice cooker. All the interviewees at Site A, who had used rice cookers in the past, considered the electric rice cooker as being excessively complicated in their operating procedure and difficult to use. The interviewees, most of whom used primarily rice cookers to cook rice, argued that not all its functions were necessary and that the excessive number of interface buttons and functions hampered easy operation. The interviewees suggested replacing the rice cooker interface design with a single-button and single-function design. Nevertheless, the interviewees regarded the rice cooker as aesthetically pleasing.

Four of the interviewees at Site B had previously used electric rice cookers. These interviewees contended that the operating procedure of the cooker was complex and difficult to memorize. Specifically, they sometimes forgot to press the button that activates the cooking command, preventing them from completing the operation. Moreover, the time programming function was complicated. The text on the interface was excessively small, and the rice cooker interface was generally complex. The interviewees, who used only rice cookers to cook rice, remarked that the number of functions was excessively high, and they required assistance from family members to use functions other than cook. The interviewees suggested replacing the interface with a single-button design or a design that starts cooking automatically after being plugged in and cuts or reduces power or after cooking is complete.

#### 4.1.2. TV Remote Controls

Table 2 lists the Site A and B interview results for the TV remote controls. The interviewees at Site A, most of whom were adept at using TV remote controls, regarded the sizes and spacing of the buttons as overly small. All the interviewees were able to read the text on the control interfaces but considered finding the positions of the volume control buttons difficult. Designs closer to the remote controls at the interviewees’ home were easier for them to understand. The interviewees primarily used TV remote controls to turn TVs on or off, adjust the volume, and select channels. The interviewees primarily identified button functions by reading text and considered the text excessively small. The interviewees suggested that the buttons, their spacings, and their text be enlarged, voice control be implemented, TV power switch, channel, and volume commands be prioritized, and the overall sizes of the controls reduced following a decrease in the number of control functions.

Most of the interviewees at Site B were experienced in using TV remote controls, but two mentioned that they usually only used the controls to turn TVs on or off and required the assistance of family members to select channels and adjust the volume. The interviewees indicated that the buttons were overly small. They found it difficult to adjust the TV volume and were worried about causing their desired screens to disappear without easy recovery because they pressed an incorrect button. The number of buttons was overly high, but the control interfaces were easier to understand if they were closer to those of the remote controls at the interviewees’ home. The interviewees used the remote controls primarily to turn TVs on or off, adjust the volume, and select channels. The text on the control interfaces was too small to read. The interviewees suggested that the buttons and text be enlarged and the control functions be simplified and prioritize TV power, channel, and volume commands. Moreover, the number of buttons should not be overly high; only the necessary functional buttons should be preserved.

### 4.2. Individual Interviews with Users with MCI

Table 3 lists the findings from the individual interviews with eight users with MCI regarding the use of the electric rice cooker and TV remote controls. Most of these interviewees were not familiar with the operating procedure of the rice cooker and considered it overly complicated to operate. Some of the interviewees learned to use them after receiving instruction but subsequently forgot how to use them, and they expressed difficulty in memorizing the operating procedure. The interviewees generally considered the “function select” buttons difficult to use and tended to directly press the functional text next to the screen. The interviewees regarded the text as too small to read. Most of the interviewees had used mainly electric rice cookers to cook rice but considered the functions provided by the test rice cooker as excessive. The interviewees provided no opinions regarding the appearance of the rice cooker, but they suggested that the text next to the screen should be directly pressable.

With the findings from the individual interviews with the users with MCI regarding the TV remote controls, all the interviewees indicated that they were able to operate the controls primarily for turning TVs on or off, selecting channels, and adjusting the volume. Most selected channels by pressing the number buttons. The interviewees considered the channel selection function difficult to operate but regarded the text on the controls large and easy to read. Regarding the appearances of the remote controls, the interviewees preferred large controls with colored buttons. They suggested that the remote controls provide only simple and necessary functions.

### 4.3. Individual Interviews with Users with Mild Dementia

Table 4 lists the findings from the individual interviews with seven users with mild dementia regarding the use of the electric rice cooker and TV remote controls. Most of the interviewees were unclear about the operating procedure and required assistance from their families to use cookers. Most considered the indirect “function select” button difficult to operate, and some were worried of forgetting to press the cooking command button. Most considered the text too small to read. Most had used rice cookers solely for cooking rice, but they considered the multiple functions provided by the rice cooker in this study as practical. The interviewees provided no opinions regarding the appearance of the rice cooker, but they suggested that the text and images on the screen be enlarged. Additionally, text that can be directly pressed next to the screen and voice control were preferrable, and a single activation button should be implemented.

With the findings from the individual interviews with seven users with mild dementia regarding the use of TV remote controls, the interviewees required assistance from their families in turning TVs on or off, selecting channels, and adjusting the volume. They could not find the volume and channel buttons because they could not read the text on the buttons. They preferred large buttons and text. Moreover, the interviewees indicated that the functions provided by the controls were excessive and might cause them to press the wrong button, and they would be unable to return to the desired screen. The interviewees used the remote controls primarily for turning on or off TVs, selecting channels, and adjusting volumes. The interviewees preferred remote controls with appearances and interface configurations similar to those of controls they used at home. Moreover, the interviewees suggested voice control for easy operation, and the functions and interfaces of the remote controls should be simplified.

### 4.4. Discussion

Some of the control participants considered the operating procedure of the electric rice cooker overly complex and difficult to memorize, and some of the participants with MCI and mild dementia were unfamiliar with the procedure. MCI affects abilities involved in performing complex daily living activities, such as using technological products [5], and dementia leads to introduce anomalies in performing such activities. The participants with mild dementia invited in this study have the symptom of obvious decline in cognitive ability. The participants might in part forget how to operate the product interface they used to use. Thus, they obviously spent more time performing the operational tasks set in this study. Besides, the results of this study revealed that most participants with mild dementia were unclear about the operating procedure and required assistance from their families to use cookers and TV remote controls. Apart from the findings above, there are not significant differences in the comparison of interview results from subjects with MCI and mild dementia. Although some of the control participants and those with MCI operated the rice cooker smoothly after instruction, they found memorizing the complex operating procedure difficult. This is consistent with findings by Schmidt and Wahl [6] that users with MCI encountered problems in their interactions with complex operating procedures. The operating interfaces and procedures of rice cookers should be simplified; for example, the single-button operation model suggested by the participants is preferrable. Generally, the control participants found the operating procedure of the rice cooker complex, and the participants with MCI and mild dementia were unfamiliar with the procedure. This indicates that interface and procedural design affect the ease with which contrololder adults and those with cognitive impairment use products at different levels. According to Brankaert and Ouden [9], users with dementia have difficulty understanding complex functional structures. According to the participants in this study, control or diagnosed as having MCI or mild dementia, the functions provided by a product should not be excessive. Therefore, simplified operating procedures and functions are the most favorable product interface design for older adults and individuals with cognitive impairment. Moreover, some of the control participants had difficulty using the rice cooker without prior experience, which plays a crucial role in the interaction between users and products. However, cognitive impairment substantially inhibited the interaction of the participants with MCI or mild dementia with the product interface despite prior experience in using rice cookers.

Some of the participants with MCI or mild dementia experienced difficulty using the “function select” button on the rice cooker interface (highlighted with a red frame in Figure 6) because it is used for indirectly selecting cooking modes (highlighted with a blue frame in Figure 6) such as for white rice and quick cooking. Notably, some of the participants with MCI or mild dementia suggested that the text on the interface be directly pressable for easy operation. Further research is required on whether a more direct operating interface design fulfills the needs of users with cognitive impairment.

The control participants had no difficulty in operating the TV remote controls because of the simple TV power switch, channel selection, and volume adjustment procedures. Some of the participants with MCI were able to operate the controls by themselves, and most of these participants used the number buttons to select channels. Some of the participants with mild dementia required assistance from their family in turning TVs on or off, selecting channels, and adjusting volume and suggested that voice control be implemented for easier operation. This revealed the anomaly in users with mild dementia in performing the operating procedures of the TV remote controls, even simple ones. For users with advanced age and cognitive impairment, the buttons on a TV remote control should be large, and the number of buttons should not be excessive; only the necessary functions should be provided, and complex operating procedures should be avoided. According to the participants with MCI or mild dementia, direct operation and voice control are design solutions worth considering.

The results revealed that the control participants considered the operating interface design of technological products primarily from the perspectives of ease of operation (e.g., button and text size), functions, and prior user experience, whereas the participants with mild dementia, who were unfamiliar with the operating procedures, preferred clear operating instructions (e.g., voice control or directly pressing text). According to the interactive design scholars Blackler and Hurtienne [19], users’ prior experience and knowledge play a critical role in intuitive interaction. However, the role of prior user experience may become diminished for users with dementia because of the deterioration of their cognitive abilities and short-term memory. For intuitive interaction between users with notable cognitive impairment and product interfaces, procedural operations should be replaced with simple operations (e.g., single-button operation), and clear messages (i.e., feedback) should be provided to present the expected functions and operation methods of products. The results from the older adults with MCI and mild dementia in this study provide practical design references for product interface design for elders.

## 5. Conclusions

This study involved two focus group interviews with 10 control older adults and individual interviews with eight individuals with MCI and seven with mild dementia regarding an electric rice cooker and three TV remote controls. The control participants stressed that the operating procedures of the products should not be excessively complex, the number of functions and buttons should not be overly high, and buttons and text should be enlarged. For the participants with MCI or mild dementia, in addition to the size and number of buttons, text size, and functions, their operation of the product interfaces was affected by the complexity of the operating procedures. The solutions recommended by the participants included interface designs for direct operation or voice control. Due to disease factors, the participants with MCI or mild dementia were interviewed individually. Although they were guided by family and caregivers during the interviews, the opinions they expressed regarding the use of the test products were limited. Furthermore, when these participants were guided or instructed in using the products, the disease factors prevented them from effectively memorizing the operating procedures. Future studies should adjust participants’ product operation time. For example, the product interface operation time for participants can be extended, or participants can be assigned to use the test products in daily living for a predetermined period (e.g., 2 weeks). Thus, participants can familiarize themselves with test products and provide opinions regarding their needs and the design of the products.

Although all the participants were experienced in using home appliances and technological products, they differed individually in their product use experience. Some of the participants had not cooked rice themselves or had relied on the assistance of family members or caregivers in using TV remote controls, and they were unable to provide opinions or suggestions on product designs during the interviews. Future studies should invite users highly experienced in operating specific products, such as older adults who use electric rice cookers regularly, to acquire more detailed interview results. In this study, usage experience was considered in selecting participants, but differences in education levels were not considered. Differences in education levels may have influenced the interview results. Future studies should consider the education levels of participants for clearer research results; specifically, the effect of education level on the operation of product interfaces by users with advanced age and cognitive impairment should be further examined. Electric rice cookers and TV remote controls are among the most prevalent daily living products. Further research should employ other types of technological products as research tools to explore different dimensions of product interface design requirements and to clarify the characteristics of the interaction between users with advanced age and cognitive impairment and product interfaces, providing a valuable reference for product design.

## Figures and Tables

**Figure 1 ijerph-19-02466-f001:**
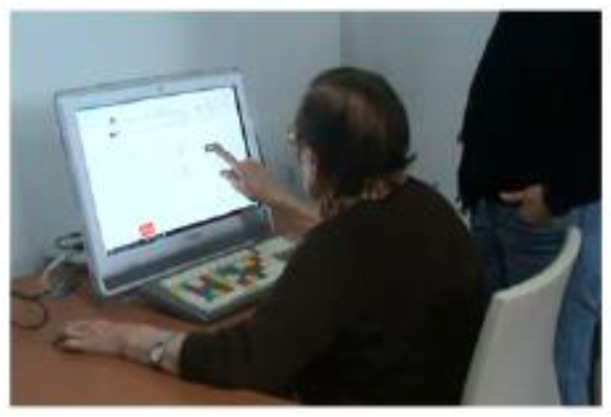
Usability test [15].

**Figure 2 ijerph-19-02466-f002:**
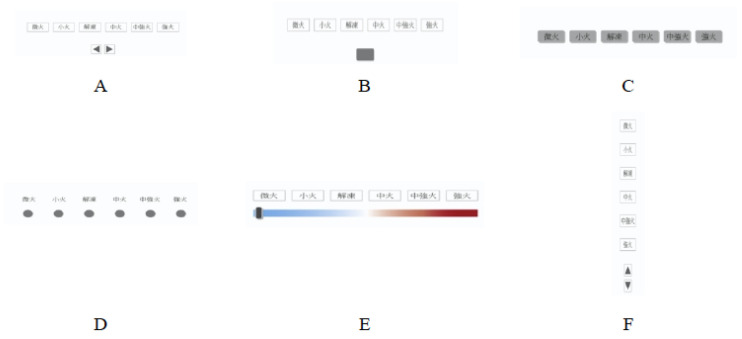
Test interfaces for adjusting cooking power [16]. Interfaces (**A**,**B**,**F**) were operated by pressing buttons at the bottom of the interface to cycle through the available power settings. Interface (**C**) was operated by directly pressing the buttons labeled with the different power settings. Interface (**D**) was operated by pressing the round buttons directly underneath the power setting labels. Finally, Interface (**E**) was operated by sliding a bar underneath the power setting labels.

**Figure 3 ijerph-19-02466-f003:**
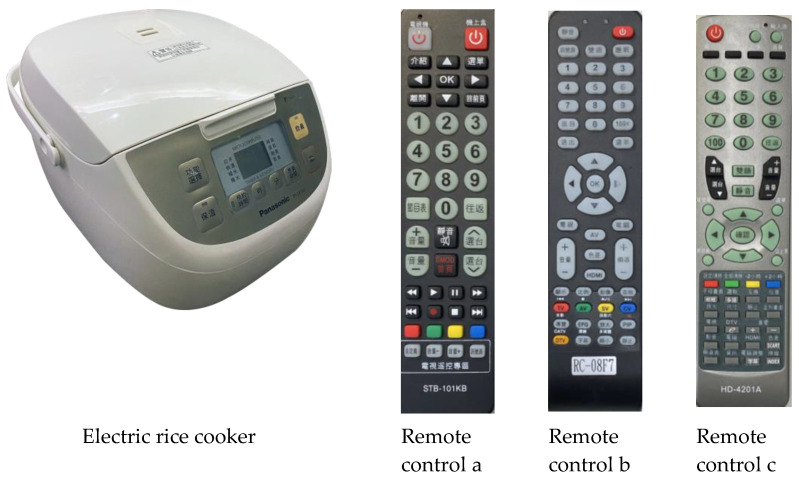
Electric rice cooker and TV remote controls used as test products.

**Figure 4 ijerph-19-02466-f004:**
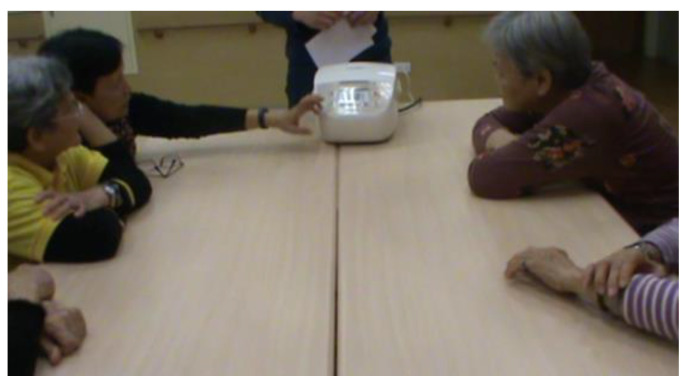
Focus group interview.

**Figure 5 ijerph-19-02466-f005:**
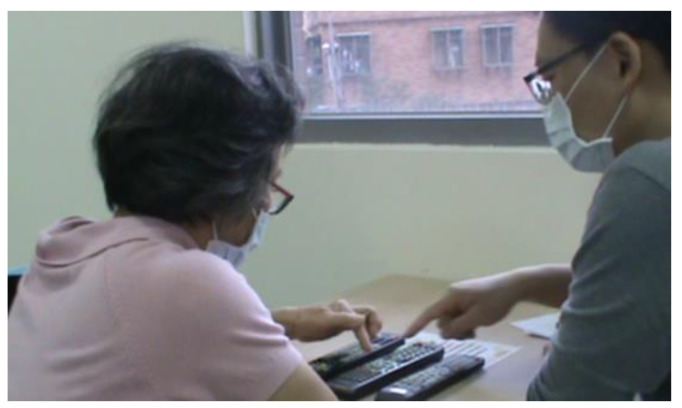
Individual interview.

**Figure 6 ijerph-19-02466-f006:**
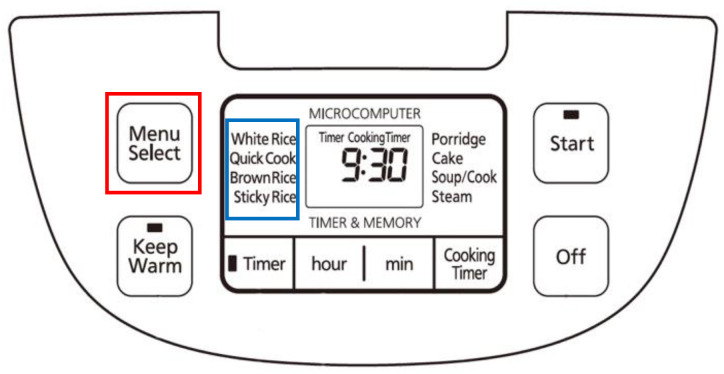
Operating interface of the rice cooker.

**Table 1 ijerph-19-02466-t001:** Sites A and B focus group interviews on the electric rice cooker.

Site A		
Item	Content	Suggestions
Operating procedure	● Complicated ● Difficult to memorize (number of procedures is too high) ● Difficult to operate without prior experience	● Single-button operation ● Single-function design
Operating interface	● Number of functional buttons is too high	
Functions	● Not all the functions are necessary ● Number of functions is too high ● Used primarily for cooking rice	
Appearance	● Aesthetically pleasing	
**Site B**		
**Item**	**Content**	**Suggestions**
Operating procedure	● Complex and difficult to memorize ● Interviewees sometime forgot to press the cooking command button ● Cooking time setting function is complicated	● Single-button operation ● Starts cooking automatically after plugging in and cuts power or preserves heat after cooking is complete
Operating interface	● Text is too small to read ● Interface is complicated	
Functions	● Number of functions is too high ● Usually used mainly for cooking rice ● Using other functions requires assistance from family members	
Appearance	● None	

**Table 2 ijerph-19-02466-t002:** Site A and B focus group interviews on TV remote controls.

Site A		
Item	Content	Suggestions
Operating procedure	None	● Buttons should be enlarged ● Button spacing should be enlarged ● Voice control should be implemented ● Text should be enlarged ● Number of functions should not be too high; only necessary functions should be kept (viz., TV switch, channel, and volume commands) ● Control sizes can be reduced by eliminating unnecessary buttons
Operating interface	● Buttons are too small and complicated ● Button spacing is too small ● Interfaces closer to those at home are easier to understand ● Volume buttons difficult to find ● Functional text unreadable ● Button positions immediately recognizable for interviewees accustomed to the operating procedure ● Text too small ● Button functions assessed primarily by reading text	
Functions	● Used primarily for turning TVs on or off, adjusting volume, and selecting channels ● Selecting channels is straightforward (using the number and up/down buttons) ● Other functions rarely used	
Appearance	● None	
**Site B**		
**Item**	**Contents**	**Suggestions**
Operating procedure	None	● Buttons should be enlarged ● Text should be enlarged ● Functions should be simplified, prioritizing TV switch, channel, and volume commands ● Number of buttons should not be too high; only the necessary functional buttons should be kept
Operating interface	● Buttons too small ● Adjusting volume is difficult ● Pressing the wrong button may cause the screen to disappear, making it difficult to recover ● Interfaces closer to those at home are easier to understand ● Number of functional buttons is too high ● Text too small	
Functions	● Used only for turning TV on or off, with other functions requiring assistance from family members to use ● Used primarily for turning TV on or off, selecting channels, and adjusting volume ● Selecting channels is straightforward through the use of the number buttons ● Other functions rarely used for the fear of pressing the wrong button and causing the screen to disappear	
Appearance	● None	

**Table 3 ijerph-19-02466-t003:** Interviews with users with MCI regarding electric rice cooker and TV remote controls.

Item	Products	Contents	Suggestions
Operating procedure	ERC	● Interviewees learned to use after receiving instructions ● Interviewees unfamiliar with the procedure ● Interviewees tended to forget the procedure and considered it difficult to memorize ● Procedure overly complicated	ERC: ● Functional text should be directly pressable
	TRC	● Interviewees able to operate the controls on their own ● Most interviewees used the number buttons to select channels	TRC: ● Functions should be simplified ● Unnecessary functions should not be provided
Operating interface	ERC	● “Function select” buttons difficult to operate ● Four of the interviewees want to press the text directly ● Text too small to read	
	TRC	● Channel selection function difficult to operate ● Large texts preferrable	
Functions	ERC	● Used primarily for cooking rice ● Number of functions too high ● Cooking time too long ● Easy to damage because of excessive functions	
	TRC	● Used primarily for turning on or of TVs, selecting channels, and adjusting volumes	
Appearance	ERC	● None	
	TRC	● Large controls preferrable ● Colored buttons preferrable	

Electric rice cooker (ERC); TV remote controls (TRC).

**Table 4 ijerph-19-02466-t004:** I Interviews with users with mild dementia regarding electric rice cooker and TV remote controls.

Item	Products	Content	Suggestions
Operating procedure	ERC	● Interviewees unfamiliar with the procedure ● Interviewees required assistance from family members to use the cooker ● Pressing the “function select” button was difficult ● Interviewees tended to forget to press the cooking command button	ERC: ● Large images preferrable ● Large text preferrable ● Voice control preferrable ● Text that can be directly pressed next to the screen preferrable ● Single-button operation preferrable
	TRC	● Interviewees required assistance from their families in turning TVs on or off ● Interviewees required assistance from their families in selecting channels ● Interviewees required assistance from their families in adjusting volume	
Operating interface	ERC	● Text on screen too small ● Indirect “function select” buttons difficult to use	
	TRC	● Larger buttons and texts were easier to use ● Interviewees unable find the volume and channel buttons ● Interviewees unable to read the text on the buttons ● Controls with large buttons and text preferrable	
Functions	ERC	● Used solely for cooking rice ● Multiple functions usable	TRC: ● Voice control preferrable ● Single-function control preferrable ● Simple interfaces preferrable
	TRC	● Number of functions too high ● Interviewees worried of being unable to return to desired screen because they pressed the wrong button ● Controls used primarily for turning TVs on or off, selecting channels, and adjusting volume	
Appearance	ERC	● None	
	TRC	● Interviewees preferred controls with an appearance similar to those they use at home	

Electric rice cooker (ERC); TV remote controls (TRC).

## Data Availability

Not applicable.

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
