# Peer review of "Interface Design for Products for Users with Advanced Age and Cognitive Impairment"

_ijerph, 2022, doi:10.3390/ijerph19042466_

Round 1
Reviewer 1 Report
The present manuscript investigates how cognitive impairment affects community-dwelling older adults' ability to utilize the rice cooker and the TV remote controller in Taiwan. While the ability to manage daily activities on their own is an essential criterion for dementia diagnosis, the study findings are important to build not only more dementia friendly communities but also dementia free communities in future.
The manuscript is overall easy to follow the logit and summarize well; however the authors can describe a little more details on the following points:
- the reasons why the authors selected the rice cooker and the TV remote controller rather than any other electric appliances such as smart phone (are they particularly important for Taiwanese older adults to maintain independent living? or are they the first devices many older adults face difficulty?)
- how the authors acquired informed consent from the participants with MCI or dementia. There should be a sentence indicating the research ethic compliance.
- the types of dementia the participants suffered from. There are different types of dementia and their symptoms vary. Please describe what types of dementia the participants were suffering from and discuss how they affect the findings.
Author Response
Point1. the reasons why the authors selected the rice cooker and the TV remote controller rather than any other electric appliances such as smart phone (are they particularly important for Taiwanese older adults to maintain independent living? or are they the first devices many older adults face difficulty?)
Response 1: With selecting the rice cooker and TV remote control, Taiwanese older adults frequently use these two electric appliances in their daily life. That is the main reason for us to decide to select the rice cooker and TV remote control as the test sample in this study.
Point2. how the authors acquired informed consent from the participants with MCI or dementia. There should be a sentence indicating the research ethic compliance.
Response 2: We carefully asked and invited every participant whether they were willing to participate the interviews. We told the participants the details the interviews if they could participate the interviews. After that, the participants signed the formal consent document.
Point3. the types of dementia the participants suffered from. There are different types of dementia and their symptoms vary. Please describe what types of dementia the participants were suffering from and discuss how they affect the findings.
Response 3: The participants with mild dementia invited in this study have the symptom of obvious decline in cognitive ability. They might in part forget how to operate the product interface they used to use. Thus, they obviously spent more time to perform the operational tasks set in this study. Besides, the results of this study revealed that most participants with mild dementia were unclear about the operating procedure and required assistance from their families to use cookers and TV remote controls. Apart from the findings above, there are not significant difference in comparison of interview results from subjects with MCI and mild dementia.
We have described the participants with mild dementia invited and discussed how they affect the findings.
(Page 11, from Line 334 to 342)

Reviewer 2 Report
The subject of advanced age is very timely and very universal. This paper may be interesting to a very wide audience: academics, policy makers, social services, etc.
I find this topic very interesting, useful and up to date in many contexts, e.g.: social, demographic, economic, administrative.
The applied research methods are adequate to the problem.
The article contains almost the appropriate structure. The article has been correctly divided into relevant sections, and their content coincides with their titles. However, there is no section devoted to the literature analysis. It should be supplemented.
Footnotes and bibliography are correctly formulated.
The language of the article is mature, correct, adequate.The work is aesthetic, presented in an interesting way.
However, some parts of the article have to be strengthened:
- add section devoted to literature review;
- in my opinion, the number of references is not sufficient (15 is not enough). Authors should add to the paper much more references. When the Authors supplement the paper with part devoted to literature review, the number of references should also increase. The authors of the paper focused very much on the part devoted to the research, but without a proper background in the literature, the effect will not be so good. Additionally, the presented literature on the subject is not very up-to-date.
- I can't find out when the survey was conducted;
- Can the authors indicate whether similar studies have been carried out in their country? Or maybe they can indicate examples of similar research in other countries? Were the conclusions similar?
- apart from the abstract, the purpose of the article/research was not specified
- no hypothesis was given;
Author Response
Point 1: add section devoted to literature review;
Response 1: We have added the Literature review Section in the manuscript.
(Page 2, Line 70)
Point 2: in my opinion, the number of references is not sufficient (15 is not enough). Authors should add to the paper much more references. When the Authors supplement the paper with part devoted to literature review, the number of references should also increase. The authors of the paper focused very much on the part devoted to the research, but without a proper background in the literature, the effect will not be so good. Additionally, the presented literature on the subject is not very up-to-date.
Response 2: We have added the related studies in the manuscript.
(Pages 3 and 4, from Line 100 to 134)
Point 3: I can't find out when the survey was conducted;
Response 3: The focus group interviews to older adults were held in May of 2020, and the individual interviews with the participants with MCI and mild dementia were held in July and August of 2020.
We have indicated the time of the survey in Subsection “2.2 Focus Group and Individual Interviews” of the manuscript.
(Page 5, Lines 187, 197 and 198)
Point 4: Can the authors indicate whether similar studies have been carried out in their country? Or maybe they can indicate examples of similar research in other countries? Were the conclusions similar?
Response 4: We have indicated the similar studies in Discussions Section of this manuscript.
(Page 11, from Line 344 to 355)
Point 5: apart from the abstract, the purpose of the article/research was not specified
Response 5: We have indicated the purpose of this study in Introduction Section of this manuscript.
(Page 2, from Line 66 to 69)
Point 6: no hypothesis was given;
Response 6: We have indicated the hypothesis of this study in Introduction Section of this manuscript.
(Page 2, from Line 62 to 65)
